# Ornidazole-Induced Liver Injury: The Clinical Characterization of a Rare Adverse Reaction and Its Implications from a Multicenter Study

**DOI:** 10.3390/biomedicines13071695

**Published:** 2025-07-11

**Authors:** Ali Rıza Çalışkan, Ilker Turan, Sezgin Vatansever, Jasmin Weninger, Emine Türkmen Şamdancı, Ayşe Nur Akatli, Elvan Işık, Esra Durmazer, Ayşenur Arslan, Nilay Danış, Hüseyin Kaçmaz, Sedat Cicek, Osman Sağlam, Dilara Turan Gökçe, Derya Arı, Sevinç Tuğçe Güvenir, Serkan Yaraş, Cumali Efe, Meral Akdoğan Kayhan, Murat Harputluoğlu, Ali Canbay, Ulus Salih Akarca, Zeki Karasu, Ramazan Idilman, Fulya Günşar

**Affiliations:** 1Department of Gastroenterology, Faculty of Medicine, Adiyaman University, Adiyaman 02040, Turkey; acaliskan@adiyaman.edu.tr (A.R.Ç.); drkacmaz02@hotmail.com (H.K.); drsedatcicek23@gmail.com (S.C.); 2Department of Gastroenterology, Faculty of Medicine, Ege University, Izmir 35040, Turkey; elvanisik80@hotmail.com (E.I.); ulusakarca@gmail.com (U.S.A.); zekikarasu@gmail.com (Z.K.); fgunsar@yahoo.com (F.G.); 3Department of Gastroenterology, Izmir Ataturk Education and Research Hospital, Izmir 35360, Turkey; sezginvatansever19@gmail.com; 4Department of Internal Medicine, Knappschaft Kliniken Universitätsklinikum Bochum, Ruhr University, 44892 Bochum, Germany; jasmin.weninger@knappschaft-kliniken.de (J.W.); ali.canbay@knappschaft-kliniken.de (A.C.); 5Department of Pathology, Faculty of Medicine, Gazi University, Ankara 06560, Turkey; esamdanci@gmail.com; 6Department of Pathology, Faculty of Medicine, Inonu University, Malatya 44000, Turkey; aysenurakatli@gmail.com; 7Department of Internal Medicine, Faculty of Medicine, Ege University, Izmir 35040, Turkey; dr.esradurmazer@gmail.com (E.D.); dr.aysenurarslan@gmail.com (A.A.); 8Department of Gastroenterology, Faculty of Medicine, Dokuz Eylül University, Izmir 35390, Turkey; nilaydanis@gmail.com; 9Department of Gastroenterology, Faculty of Medicine, Inonu University, Malatya 44000, Turkey; ossag03@gmail.com (O.S.); mharputluoglu@hotmail.com (M.H.); 10Department of Gastroenterology, Ankara Bilkent City Hospital, Ankara 06800, Turkey; dilaraturan89@yahoo.com (D.T.G.); deryaari81@hotmail.com (D.A.); akdmeral@yahoo.com (M.A.K.); 11Department of Gastroenterology, Faculty of Medicine, Ankara University, Ankara 06100, Turkey; stugcekarayel@gmail.com (S.T.G.); ramazan.idilman@medicine.ankara.edu.tr (R.I.); 12Department of Gastroenterology, Faculty of Medicine, Mersin University, Mersin 33343, Turkey; drserkanyaras@gmail.com; 13Department of Gastroenterology, Faculty of Medicine, Harran University, Sanlıurfa 63050, Turkey; drcumi21@hotmail.com

**Keywords:** drug-induced liver injury, ornidazole-induced liver injury, hepatitis, liver transplantation, corticosteroid therapy

## Abstract

**Background and Aims:** Ornidazole, a nitroimidazole antibiotic, is widely used for protozoal and anaerobic infections and is generally considered safe. However, ornidazole-induced liver injury (OILI) is an underrecognized yet potentially severe adverse reaction. This multicenter study aims to characterize the clinical features, histopathology, and outcomes of OILI to improve the awareness and management of this rare entity worldwide. **Methods:** We conducted a retrospective analysis of 101 patients with OILI from eight tertiary centers between 2006 and 2023. Cases were included based on liver enzyme elevations temporally linked to ornidazole and the exclusion of other causes. Causality was assessed using the Roussel Uclaf Causality Assessment Method (RUCAM) score. Clinical data, laboratory parameters, autoantibody profiles, histology, treatments, and outcomes were evaluated. **Results:** OILI was classified as highly probable in 42.6% of cases (n = 43), probable in 51.5% of cases (n = 52), and possible in 5.9% (n = 6) of cases. The predominant pattern was acute hepatocellular injury (83.2%) (n = 84). Autoimmune-like hepatitis occurred in 5% of cases (n = 5), with ANA positivity in 16.8% of cases (n = 17). Corticosteroids were used in 24.8% of cases (n = 25) and were associated with higher ANA positivity and a 20% (n = 5) relapse rate post-discontinuation. Recovery was achieved in 87.7% of cases (n = 88), while 7.9% of cases (n = 8) required liver transplantation and 4% (n = 4) died. **Conclusions:** Ornidazole can cause serious idiosyncratic liver injury, including autoimmune phenotypes, and should be considered in the differential diagnosis of acute hepatitis. Given the notable risk of liver failure and death, early recognition, drug discontinuation, and close monitoring are essential. In select cases, corticosteroids and plasmapheresis may be beneficial, though the evidence remains limited.

## 1. Introduction

Drug-induced liver injury (DILI) is a growing global cause of acute liver dysfunction and failure [1,2]. It can result from a wide range of medications and supplements. A recent meta-analysis estimated an incidence of ~5 cases per 100,000 person-years, with rates rising since 2010 [3]. DILI accounts for 3–5% of jaundice-related hospitalizations and is the leading cause of acute liver failure in many Western countries [4]. Idiosyncratic DILI is typically dose-independent, unpredictable, and affects only a minority of exposed individuals. It usually manifests as acute hepatocellular injury within 5–90 days [5]. Causative agents vary by region: antibiotics are the most common trigger in Western countries, while herbal remedies and anti-tubercular drugs dominate in Asia [6].

Ornidazole, a synthetic 5-nitroimidazole antimicrobial, is used to treat protozoal and anaerobic infections. Preferred for its broad spectrum and tolerability, it is widely used in countries such as Turkey, India, China, and parts of Europe [7]. Indications include amoebic dysentery, trichomoniasis, bacterial vaginosis, and dental infections [8,9]. While common side effects are mild (e.g., nausea and a metallic taste), ornidazole-induced liver injury (OILI) is extremely rare [10,11] and can be underrecognized or misdiagnosed. Some OILI cases resemble autoimmune hepatitis or severe cholestatic injury leading to liver failure [10]. In Turkey, ornidazole represents a commonly utilized antimicrobial agent in gynecological and dental practice. This medication is specifically prescribed for treating anaerobic infections within these medical specialties.

We present an analysis of 101 OILI cases, detailing clinical features, outcomes, and causality assessments. Our goal is to raise awareness of ornidazole as a potential DILI agent, highlight diagnostic and management strategies, and promote prevention.

## 2. Methods

We conducted a retrospective study on patients who developed liver injury temporally linked to ornidazole therapy, identified via electronic health records and pharmacovigilance reports at eight university hospitals in Turkey (Adıyaman University, Ege University, Kâtip Çelebi University, Inonu University, Harran University, Ankara Bilkent City Hospital, Mersin University, and Ankara University) between December 2006 and November 2023. The inclusion criteria were as follows:

Confirmed ornidazole use;

Liver enzyme elevation (alanine aminotransferase (ALT) ≥ 5× upper limit of normal (ULN); alkaline phosphatase (ALP) ≥ 3× ULN; or ALT ≥ 3× ULN + bilirubin ≥ 2× ULN) during or shortly after therapy;

No alternative cause of liver injury [12].

Patients with pre-existing liver diseases (e.g., alcohol-related, viral, metabolic, vascular, or congenital causes) were excluded to ensure drug-induced etiology. Ethical approval was granted by the Adıyaman University Ethics Committee (protocol number: 2023/3-1).

We recorded demographics, latency, corticosteroid use, hospital stay, and recovery times. Laboratory data included ALT, ALP, aspartate aminotransferase (AST), international normalized ratio (INR), and total/direct bilirubin. The R ratio [ALT/ULN) ÷ (ALP/ULN)] was calculated to classify injuries as hepatocellular (R ≥ 5), cholestatic (R ≤ 2), or mixed (2 < R < 5) according to international DILI criteria [12]. Causality was assessed via the Roussel Uclaf Causality Assessment Method (RUCAM) score (≥8 = highly probable; 6–7 = probable; 3–5 = possible; ≤2 = unlikely; and ≤0 = relationship with the drug excluded) [13]. Study participants were selected based on Rucam causality assessment scores, including only cases classified as highly probable, probable, or possible for drug-induced hepatotoxicity. Serologic and autoimmune markers (viral hepatitis like HAV, HBV, HCV, and HEV and antinuclear antibodies (ANA) ≥ 1:40) and metabolic evaluations (e.g., Wilson’s disease) were reviewed. All patient data were de-identified.

Liver biopsies, when available, were interpreted by a blinded expert pathologist for features suggestive of DILI or autoimmune overlaps (e.g., interface hepatitis, cholestasis, necroinflammation, and fibrosis) and correlated with clinical patterns. Representative histologic images were documented (Figure 1).

Outcomes included recovery (symptom resolution and laboratory normalization <3 months), chronic DILI (>6 months of abnormal laboratories), liver transplantation, and death. Treatment regimens (e.g., corticosteroids and plasmapheresis) and outcomes were noted. Patients with autoimmune hepatitis scores ≥6 were monitored; immunosuppression was tapered after one year. Recurrence classified the case as ornidazole-induced autoimmune hepatitis; the absence of recurrence indicated ornidazole-induced autoimmune-like hepatitis [12].

Data were analyzed using IBM SPSS Statistics for Windows, Version 26.0 (IBM Corp., Armonk, NY, USA). Descriptive stats included frequencies, percentages, means, standard deviations (SDs), medians, minimum and maximum values, interquartile ranges (IQRs), and 25th–75th percentiles (Q1-Q3). Normality was assessed via histograms, Q-Q plots, skewness, kurtosis, and the Shapiro–Wilk test. Non-normally distributed data were compared using the Mann–Whitney U test (2 groups) or Kruskal–Wallis test (>2 groups), with the Bonferroni–Dunn also used for post hoc analysis. Categorical variables were analyzed using Pearson’s χ^2^ if fewer than 20% of expected cell counts were below a value of five; otherwise, Fisher’s exact test was applied, with Bonferroni correction for pairwise comparisons. Correlations between numerical variables were evaluated using Spearman’s rank correlation test, as the normality assumption was not met. Statistical significance was set at *p* < 0.05.

## 3. Results

A total of 101 patients met the inclusion criteria [mean age: 46 years; range: 19–83; 80.2% female (n = 81)]. Table 1 summarizes patient characteristics. The mean latency from ornidazole initiation to liver test abnormalities was 16.9 ± 8.3 days (median: 15; range: 7–60). The median hospital stay was 13 days (range: 1–52). Among the 88 patients included, the mean time to improvement was 62.7 days (range: 14–190).

R score patterns were hepatocellular in 83.2% of cases (n = 84), mixed in 13.9% of cases (n = 14), and cholestatic in 3.0% of cases (n = 3). ANA was positive at 16.8% (n = 17) (mean titer: 1:240). Ornidazole was discontinued in all cases; 5.9% (n = 6) of cases were re-challenged. The RUCAM assessment rated causality as highly probable in 42.6% of cases (n = 43), probable in 51.9% of cases (n = 52), and possible in 5.9% of cases (n = 6), primarily due to timing (onset 5–15 days into therapy), a lack of alternative causes, and response to de-challenging.

Final diagnoses included OILI in 90.1% of cases (n = 91), ornidazole-induced autoimmune-like hepatitis in 5.0% of cases (n = 5), and ornidazole-induced autoimmune hepatitis in 5.0% (n = 5) of cases. Patients with autoimmune hepatitis scores ≥6 were monitored; immunosuppression was tapered after one year. Recurrence led to the case being classified as ornidazole-induced autoimmune hepatitis; the absence of recurrence indicated ornidazole-induced autoimmune-like hepatitis. This was based solely on a relapse after steroid withdrawal. No other criterion was used in this regard. Recovery occurred in 87.7% of cases (n = 88) overall (females: 87.7%; males: 85.0%; *p* = 0.999). Liver transplantation was required in 7.9% of cases (n = 8) (all females; *p* = 0.351). One patient developed chronic DILI (1.0%), and four died (4.0%; mortality rate male (n = 2): 10.0%, female (n = 2): 2.4%; *p* = 0.186). Plasmapheresis was used in 17.8% of cases (n = 18).

Table 2 details the peak lab values. Corticosteroids were administered in 24.8% of cases (n = 25) (see Table 3). The hospitalization period was slightly longer in steroid-treated patients (mean: 16.6 vs. 14.7 days; *p* = 0.202), with higher but not statistically significant recovery rates (96.0% vs. 85.5%; *p* = 0.285) and longer recovery times (mean: 72 vs. 59.2 days; *p* = 0.067). ANA positivity was significantly higher in steroid users (36% vs. 10.5%; *p* = 0.011).

Corticosteroids were used for a mean of 67.3 days (median: 43). Duration by sex is given as follows: 64.1 days (females) vs. 91.0 (males; *p* = 0.257). Clinical relapses occurred in 20% of cases (n = 5) after steroid discontinuation. The ANA positivity rate was significantly higher in corticosteroid users (36%, n = 9) than in non-users (10.5%, n = 8; *p* = 0.011). The liver transplantation rate was 4% (n = 1) in corticosteroid users and 9.2% (n = 7) in non-users (*p* = 0.675). No deaths or chronic cases were seen in the steroid group, whereas in non-users, four deaths (5.3%) and one case of chronicity (1.3%) were observed.

A liver biopsy was performed in 47 patients (47%) due to diagnostic uncertainty (Figure 1 and Figure 2). Histology showed pericentral hepatocyte necrosis, lobular/interface activity, lymphocytic infiltration with eosinophils and plasma cells, single-cell necrosis, Councilman bodies, and hepatocyte ballooning, the findings of which are consistent with DILI. No significant fibrosis was noted. One case showed massive necrosis with lymphoplasmacytic infiltrates (Figure 3).

**Figure 1 biomedicines-13-01695-f001:**
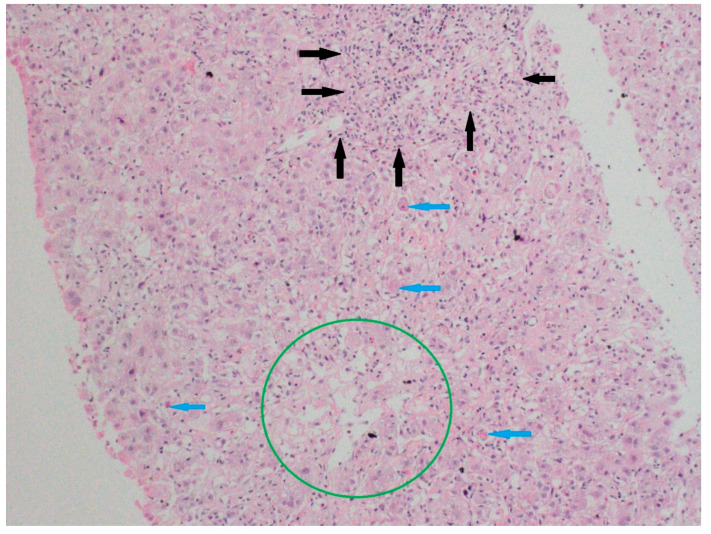
Specimens from liver biopsy. Necrosis in pericentral hepatocytes (Zone 3) (green circle); severe portal inflammation with marked interface activity (black arrows); and focal spotty necrosis in the lobular area (blue arrows). Hematoxylin–eosin; original magnification ×100.

## 4. Discussion

This multicenter case series highlights severe liver injury linked to ornidazole, which is a rarely recognized hepatotoxic antimicrobial. Patients developed acute liver injury shortly after starting ornidazole (mean latency: 16.9 days) in accordance with the existing literature, with improvement upon cessation, meeting established DILI criteria [5,9,14]. In its 2023 update, the American Association for the Study of the Liver (AASLD) recommended evaluating all drugs taken within 180 days prior to hospitalization when assessing for DILI. Key diagnostic features included the timing of onset, recovery post-discontinuation, recurrence upon re-exposure, and known hepatotoxic potential [15].

Ornidazole’s hepatotoxicity, first reported in the early 2000s, remains uncommon. Tabak et al. identified only six cases at the time [11]. Subsequent reports, mainly involving regions where ornidazole was widely used, resulted in the description of two primary phenotypes: (1) hepatocellular injury with jaundice and pruritus, and (2) autoimmune-like hepatitis marked by elevated ALT, interface hepatitis, and steroid responsiveness [9,10,16,17]. Both types were seen in our cohort. Pathogenesis is unclear but likely immune-mediated, similar to other idiosyncratic DILI cases. The presence of autoantibody-negative immune hepatitis, as observed here and elsewhere [10], suggests a possible immune tolerance disruption akin to minocycline- or nitrofurantoin-induced autoimmune hepatitis [18]. Cholestatic injury may stem from bile canaliculi dysfunction caused by ornidazole or its metabolites. Notably, one report described severe cholestatic failure from concurrent ornidazole and fluconazole use, pointing to potential drug interactions [9].

Kosar et al. documented the first case of ornidazole-induced autoimmune hepatitis, which recurred after re-exposure despite previous corticosteroid-induced remission [19]. A separate study of eight similar cases explored whether ornidazole triggers or unmasks autoimmune hepatitis. All but one patient was treated with prednisolone (30 mg/day) and azathioprine (50 mg/day), following a standardized two-year regimen. No cases progressed to acute liver failure, supporting the conclusion that ornidazole can trigger autoimmune hepatitis [20]. ANA positivity was observed in 17 patients (mean titer 1:240) and was significantly higher among corticosteroid users (36%, n = 9) compared to non-users (10.5%, n = 8; *p* = 0.011).

Recovery rates were high: 96.0% (n = 24) achieved recovery among corticosteroid-treated patients, and 85.5% (n = 65) achieved recovery in untreated patients, with no significant difference. Recovery time averaged 59.2 days in untreated cases and was about 12 days longer in those receiving steroids. Autoimmune hepatitis or autoimmune-like hepatitis was diagnosed in 5% (n = 5) of cases, while 90.1% showed typical acute hepatitis features. The prevalence of drug-induced autoimmune-like hepatitis (DI-ALH) in DILI cohorts ranged from 3% to 8.8% [21,22,23], and one analysis found that 9.2% (24 of 261) of autoimmune hepatitis cases were later identified as DI-ALH [24]. Patients who stopped ornidazole recovered in 2–3 months; those on corticosteroids required longer periods of 3–6 months [25]. Importantly, corticosteroids did not increase the risk of acute liver failure (ALF) and were linked to liver enzyme normalization in severe DILI [26].

Previous case reports of ornidazole-induced hepatitis showed full recovery after discontinuation, with liver enzymes normalizing in about three weeks and overall stabilization achieved within six [7,27,28,29]. In our cohort, recovery occurred in 87.7% (n = 71) of females and 85.0% (n = 17) of males (not statistically significant). The average recovery times were 61.5 days for females and 67.4 days for males (*p* = 0.44). Liver failure often began with worsening jaundice (mean peak bilirubin: 16.0 mg/dL) and elevated INR (mean peak: 1.87), underscoring the need for close monitoring.

## 5. Management and Outcomes

There is no proven antidote for idiosyncratic DILI [18]; management relies on discontinuing the offending drug and initiating supportive care. Most patients in our study improved after stopping ornidazole, with a mean recovery time of 62.7 days and no significant gender difference. A review of 22 randomized controlled trials found that interventions like silymarin, bicyclol, isoglycyrrhizinate, and N-acetylcysteine had inconsistent outcomes, with no therapy altering DILI’s course [18]. Corticosteroids are the exception and are recommended by the ACG for suspected drug-induced autoimmune hepatitis or hypersensitivity features (e.g., a fever, rash, or eosinophilia) [30,31,32]. Our steroid-treated patients showed rapid improvement, supporting earlier reports [18]. Corticosteroids were used in 24.8% (n = 25) of cases, with 20.0% (n = 5) experiencing relapse post-discontinuation. If liver function fails to improve or worsens within 1–2 weeks after drug withdrawal, high-dose corticosteroids with gradual tapering may be warranted for up to six months [33]. Among the 25 patients who received corticosteroid therapy, antinuclear antibody (ANA) testing yielded negative results in 16 cases (64%). Corticosteroid treatment was initiated for these patients due to persistent jaundice and pruritus.

In contrast, a retrospective study of 361 ALF patients found that corticosteroids did not improve survival in DILI, and the outcomes were worse in severe liver injury. Among 131 DILI-related ALF patients, survival was 69% with and 66% without corticosteroids [34].

Most DILI is acute and self-limiting, though some cases progress to chronic injury or require a transplant. One patient in our cohort developed chronic DILI; eight underwent liver transplantation (4.0% of steroid users vs. 9.2% of non-users), and 4.0% (n = 4) in the non-steroid group died. Registry data show how liver-related mortality/transplantation rates vary: 4% in Spain, 6% in the US, and up to 15% in Korea [35].

TPE, a non-selective apheresis technique, was used in 17.8% of our patients. TPE is utilized in conditions with unclear pathogenic mechanisms or when more selective apheresis therapies are unavailable [36]. TPE was initiated in these patients due to persistent jaundice and pruritus. Hospital stay duration was similar between corticosteroid-treated and untreated groups, at about two weeks, which is consistent with expected recovery timelines.

Six patients experienced recurrence after accidental re-exposure, underscoring the importance of documenting causative drugs and educating patients. Reporting to pharmacovigilance systems is essential to refine ornidazole’s safety profile.

Clinicians should weigh the rare risk of severe hepatotoxicity when prescribing ornidazole. Early detection and prompt discontinuation are key to preventing liver failure [10]. Referral to a hepatologist is advised for patients with bilirubin levels >3 mg/dL or coagulopathy, as these patients need close monitoring and possibly hospitalization. If autoimmune features are evident, corticosteroid therapy may be lifesaving, as shown in our series.

This study provides a comprehensive look at ornidazole-induced liver injury, combining original clinical data with an up-to-date literature review. This study presents the largest reported cohort of ornidazole DILI cases, supporting causality and management strategies. Limitations include the retrospective design, which carries inherent biases in case identification, the small sample size reflecting the rarity of the event, and the lack of genetic data. Future studies should explore HLA genotypes, such as those with flucloxacillin (HLA-B*5701) [37]. Nonetheless, the study’s clinical consistency and alignment with published data strengthen our conclusions.

## 6. Conclusions

Ornidazole-induced liver injury, though rare, can be severe, ranging from acute hepatitis to autoimmune-like cases requiring transplantation. Our study, the largest cohort reported to date, underscores the need for vigilance even with commonly used drugs. Early recognition and prompt withdrawal are critical to prevent serious outcomes. The primary limitations of this study include its retrospective design, potential underreporting, and lack of standardization in treatment protocols. Ongoing research is needed to uncover underlying mechanisms and risk factors, enabling more targeted prevention and treatment strategies in addition to corticosteroids and plasmapheresis.

## Figures and Tables

**Figure 2 biomedicines-13-01695-f002:**
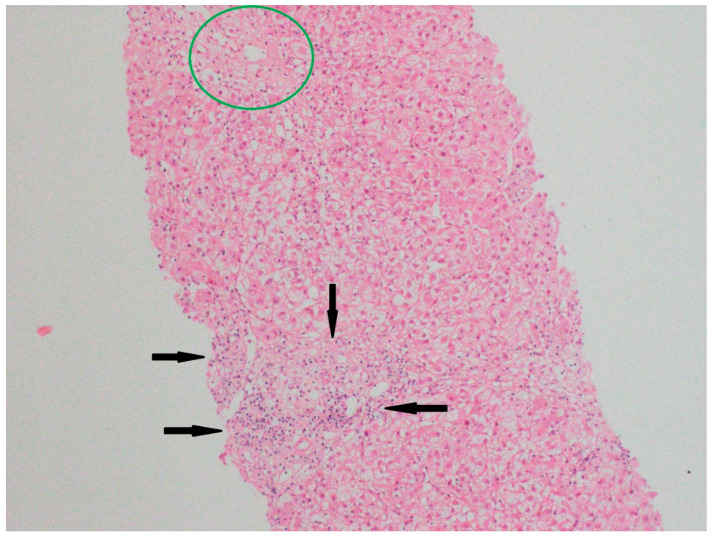
Specimens from liver biopsy. Necrosis in pericentral hepatocytes (Zone 3) (green circle); mild portal inflammation (black arrows); and hydropic degeneration of hepatocytes. Hematoxylin–eosin; original magnification ×100.

**Figure 3 biomedicines-13-01695-f003:**
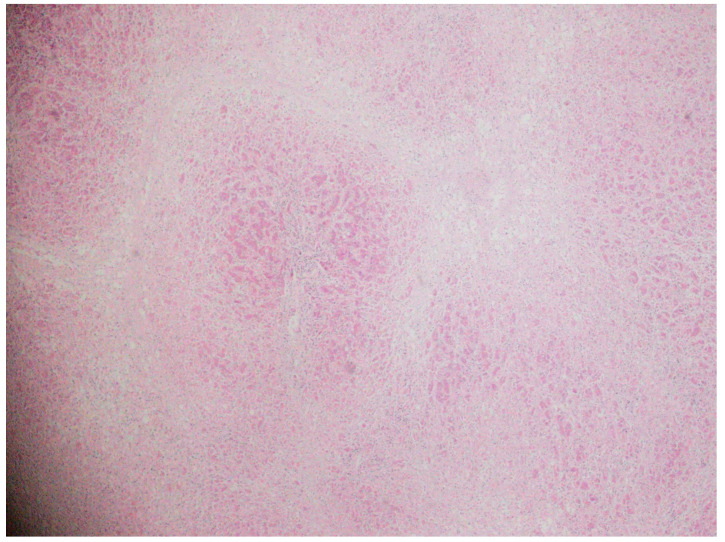
Specimens from a liver biopsy before liver transplantation. Necrosis (pale areas) in pericentral (Zone 3) and midzonal (Zone 2) hepatocytes, with portal inflammation and partially intact periportal hepatocytes (dark pink areas). Hematoxylin–eosin; original magnification ×100.

**Table 1 biomedicines-13-01695-t001:** Descriptive analysis results of ornidazole use cases.

Variable		n	%	Mean (SD)	Median (Min–Max)
**Sex**	Female	81	80.2%		
	Male	20	19.8%		
**Age (Years)**		101		46.01 (12.27)	46 (19–83)
**R Score (ALT/ULN)/(ALP/ULN)**		101		13.93 (10.92)	11.6 (0.72–54)
**R Pattern**	Hepatocellular	84	83.2%		
	Mixed	14	13.9%		
	Cholestatic	3	3.0%		
**RUCAM Score Classification**	Highly probable	43	42.6%		
	Probable	52	51.5%		
	Possible	6	5.9%		
**Latency Period (Days)**		101		16.91 (8.34)	15 (7–60)
**Hospitalization Period (Day)**		101		15.17 (9.55)	13 (1–52)
**Time to Improvement (Days)**	All	88		62.66 (31.92)	60 (14–190)
**ANA Positivity**		17	16.8%	1/240 (187.05)	1/160 (1/80–1/640)
**Rechallenge**	Yes	6	5.9%		
**Plasmapheresis**	Yes	18	17.8%		
**Final Diagnosis**	Ornidazole-induced acute hepatitis	91	90.1%		
	Ornidazole-induced autoimmune-like hepatitis	5	5.0%		
	Ornidazole-induced autoimmune hepatitis	5	5.0%		
**Outcome**	Recovery (all)	88	87.7%		
	Female	71	87.7%		
	Male	17	85.0%		
	Liver transplantation (all female)	8	7.9%		
	Chronicity	1	1.0%		
	Death (all)	4	4.0%		
	Female	2	2.4%		
	Male	2	10%		

**Table 2 biomedicines-13-01695-t002:** Peak laboratory values (n = 101).

Parameter	Mean (SD)	Median (Min–Max)
**ALT Peak Value (U/L)**	1304.31 (826.09)	1261 (112–5045)
**AST Peak Value (U/L)**	1243.32 (893.21)	1001 (55–5818)
**ALP Value at Admission (U/L)**	209.48 (113.17)	173 (55–770)
**ALP Peak Value (U/L)**	223.77 (122.05)	188 (55–793)
**Total Bilirubin Peak Value (mg/dL)**	16.01 (10.36)	15.6 (0.5–42)
**Direct Bilirubin Peak Value (mg/dL)**	11.83 (7.78)	12 (0.17–30)
**INR Peak Value**	1.87 (1.28)	1.5 (0.9–8.1)

**Table 3 biomedicines-13-01695-t003:** Comparison of corticosteroid use in treatment.

		Corticosteroid Use						
		Yes			No			
		n	Mean (SD)	Median (Q1–Q3)	n	Mean (SD)	Median (Q1–Q3)	*p* Value
**Hospital Stay (Days)**		25	16.56 (9.17)	15 (12–21)	76	14.71 (9.68)	13 (8–18)	0.202
**Recovery**	Yes	24	96.0%		65	85.5%		0.285
	No	1	4.0%		11	14.5%		
**Recovery Time** **(Days)**		24	72 (35.02)	69.5 (45–86)	65	59.22 (30.26)	56 (36–70)	0.067
**Relapses After Corticosteroid Discontinuation**	Yes	5	20.0%		0	0.0%		
	No	20	80.0%		0	0.0%		
**Liver Transplantation**	Yes	1	4.0%		7	9.2%		0.675
	No	24	96.0%		69	90.8%		
**Outcome**	Recovery	24	96.0%		64	84.2%		0.664
	Liver transplantation	1	4.0%		7	9.2%		
	Chronicity	0	0.0%		1	1.3%		
	Death	0	0.0%		4	5.3%		
**ANA**	Negative	16	64.0%		68	89.5%		0.011
	Positive	9	36.0%		8	10.5%		

## Data Availability

The data supporting this study’s findings are available on request from the corresponding author. The data are publicly available.

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
