# Peer review of "Ornidazole-Induced Liver Injury: The Clinical Characterization of a Rare Adverse Reaction and Its Implications from a Multicenter Study"

_biomedicines, 2025, doi:10.3390/biomedicines13071695_

Round 1

Reviewer 1 Report

Comments and Suggestions for Authors

The authors have analyzed a cohort of patients with hepatotoxicity attributed to ornidazole, a agent that has not been approved in the US. I have several comments to assist in making this the definitive review on the hepatotoxicity of ornidazole: 

  1. since the diagnosis did not allow for alternative causes of OILI, the authors should delete the 6 patients where the causality assessment was only possibly related, and recalculate the results for those considered probably and highly likely related.
  2. how many patients were treated with ornidazole  and had elevated liver tests  in their centers but  did not have OILI diagnosed as probably related or higher? Can the authors provide an overall estimate of prescription usage for ornidazole in their centers in order to estimate a rough incidence of OILI that is probably related ?  
  3. The authors need to be consistent when using the number of patients and the percentage of the cohort. The tables provide only numbers of patients but the text often uses just the percentage. I suggest the number be used followed by the percentage in parentheses in the text and tables. 
  4. The clinical and histologic definition of autoimmune-like injury (n=5) and autoimmune-induced injury (n=5) needs to be more fully described in the results in order to better understand how the two groups were differentiated. Was it based just on a relapse after steroid withdrawal or other criteria. These should be clearly stated.
  5. Table 1 should also include the major  treatment indications for the cohort and the mean duration of therapy. The results should comment on how many cases occurred while ornidazole was still being given (especially in cases of long-term use) as opposed to how many occurred after the course of therapy ended. 
  6. Table 2 does not mention the actual number of patients who met Hy's Law (ALT >3X and total bilirubin >2X ULN) and developed hepatocellular jaundice  (R value >5) or mixed or cholestatic jaundice (also based on R values).  I noticed on page 9 of the discussion that the term "cholestasis with jaundice" was used - but this may be an inaccurate description  since the majority of cases were hepatocellular in nature.  The term cholestasis  is defined by elevated alkaline phosphatase with R values <2. Some of these cases may have had jaundice, but the authors should not confuse the presence of hyperbilirubinemia with "cholestasis" . If the majority of cases with jaundice had an R value > 5 then this should be clearly stated as being hepatocellular jaundice. 
  7. page 10 states that steroids were administered to 24 patients. The authors need to define what the indication for steroid use was for the 14 patients who did not present with autoimmune features. Was this based on the height of the bilirubin, or symptoms, etc? 
  8. Similarly, the authors need to state the indications for use of TPE in the 18 patients who received apheresis.
  9. A large percentage of patients underwent liver biopsy - and this represents a unique opportunity for the authors to more fully describe the histologic features  and correlate them with the peak liver enzymes , the presence or absence of autoantibodies and the outcomes. A table describing the details of these individual patients would greatly strengthen the manuscript. 
  10. A comparison of the hepatic findings and outcomes seen with ornidazole  compared to other agents in the class would also be a very welcome addition in the discussion. 

Author Response

Reviewer 1:

Comments and Suggestions for Authors

The authors have analyzed a cohort of patients with hepatotoxicity attributed to ornidazole, a agent that has not been approved in the US. I have several comments to assist in making this the definitive review on the hepatotoxicity of ornidazole: 

  1. Since the diagnosis did not allow for alternative causes of OILI, the authors should delete the 6 patients where the causality assessment was only possibly related and recalculate the results for those considered probably and highly likely related.

Response: The study centers comprised university hospitals across Turkey, the majority of which function as liver transplantation centers. The RUCAM scores for six patients initially classified as "possible" were systematically recalculated and verified; however, the scores remained unchanged upon reassessment. Given that all potential etiologies of acute hepatitis were comprehensively excluded through differential diagnosis, these patients were categorized as having OILI. The primary aim of our study was to demonstrate that OILI can induce toxic hepatitis even when the RUCAM score indicates only a "possible" causal relationship. If you find our rationale satisfactory, we respectfully request that this section of the manuscript remain unmodified.

  1. How many patients were treated with ornidazole and had elevated liver tests in their centers but did not have OILI diagnosed as probably related or higher? Can the authors provide an overall estimate of prescription usage for ornidazole in their centers in order to estimate a rough incidence of OILI that is probably related? 

Response: I am the secretary of the Turkish Association of Liver Diseases, Drugs and Toxic Hepatitis Working Group. We update our records at regular intervals. Our institutional analysis revealed that 46,000 boxes of ornidazole were prescribed over a 10-year period, providing substantial exposure data for this investigation. According to the database of the Turkish Liver Research Association Toxic Hepatitis Study Group, ornidazole was identified as the second most prevalent cause of drug-induced liver injury, following amoxicillin-clavulanic acid. Similarly, in a concurrent study conducted by our research team evaluating the etiological factors of acute hepatitis at our institution, ornidazole was likewise determined to be the second most common cause of toxic hepatitis after amoxicillin-clavulanic acid. Both studies are currently in preparation for manuscript submission. Ornidazole is predominantly prescribed by gynecologists and dentists for the treatment of anaerobic bacterial infections.

  1. The authors need to be consistent when using the number of patients and the percentage of the cohort. The tables provide only numbers of patients, but the text often uses just the percentage. I suggest the number be used followed by the percentage in parentheses in the text and tables. 

Response: In the text and tables, numbers n are written in parentheses.

  1. The clinical and histologic definition of autoimmune-like injury (n=5) and autoimmune-induced injury (n=5) need to be more fully described in the results in order to better understand how the two groups were differentiated. Was it based just on a relapse after steroid withdrawal or other criteria. These should be clearly stated.

Response: Patients with autoimmune hepatitis scores ≥6 were monitored; immunosuppression was tapered after one year. Recurrence classified the case as ornidazole-induced autoimmune hepatitis; absence of recurrence indicated Ornidazole-induced autoimmune-like hepatitis. It was based solely on a relapse after steroid withdrawal. No other criterion was used in this regard.

  1. Table 1 should also include the major treatment indications for the cohort and the mean duration of therapy. The results should comment on how many cases occurred while ornidazole was still being given (especially in cases of long-term use) as opposed to how many occurred after the course of therapy ended.

Response: Ornidazole is an antibiotic that can be prescribed by all doctors in Türkiye. However, due to its anaerobic effect, it is prescribed especially by gynecologists and dentists. We do not currently know the exact number of ornidazole induced liver injuries in Türkiye. I am the secretary of the Turkish Association of Liver Diseases, Drugs and Toxic Hepatitis Working Group. We update our records at regular intervals. Ornidazole is the second most common cause of toxic hepatitis. Unfortunately, I could not add treatment indications and treatment durations to Table 1.

  1. Table 2 does not mention the actual number of patients who met Hy's Law (ALT >3X and total bilirubin >2X ULN) and developed hepatocellular jaundice (R value >5) or mixed or cholestatic jaundice (also based on R values).  I noticed on page 9 of the discussion that the term "cholestasis with jaundice" was used - but this may be an inaccurate description since the majority of cases were hepatocellular in nature.  The term cholestasis is defined by elevated alkaline phosphatase with R values <2. Some of these cases may have had jaundice, but the authors should not confuse the presence of hyperbilirubinemia with "cholestasis”. If the majority of cases with jaundice had an R value > 5 then this should be clearly stated as being hepatocellular jaundice.

Response: R pattern was added to Table 1. There was a misunderstanding. The cholestatic pattern was corrected as hepatocellular.

  1. page 10 states that steroids were administered to 24 patients. The authors need to define what the indication for steroid use was for the 14 patients who did not present with autoimmune features. Was this based on the height of the bilirubin, or symptoms, etc.? 

Response: Among the 25 patients who received corticosteroid therapy, antinuclear anti-body (ANA) testing yielded negative results in 16 cases (64%). Corticosteroid treatment was initiated in these patients due to persistent jaundice and pruritus.

  1. Similarly, the authors need to state the indications for use of TPE in the 18 patients who received apheresis.

Response: TPE was initiated in these patients due to persistent jaundice and pruritus.

  1. A large percentage of patients underwent liver biopsy - and this represents a unique opportunity for the authors to more fully describe the histological features and correlate them with the peak liver enzymes, the presence or absence of autoantibodies and the outcomes. A table describing the details of these individual patients would greatly strengthen the manuscript.

Response: This study represents the first comprehensive multicenter evaluation of patients with ornidazole-induced liver injury (OILI). Our research group has developed an extensive research agenda focused on OILI, with particular emphasis on investigating HLA typing and other genetic polymorphisms. Future studies will specifically examine genetic differences between patients who develop OILI and those who receive ornidazole therapy without experiencing hepatotoxicity.

Subsequent studies in our research program will encompass detailed evaluation of liver histopathology alongside human leukocyte antigen (HLA) genotyping and analysis of other relevant genetic polymorphisms.

  1. A comparison of the hepatic findings and outcomes seen with ornidazole compared to other agents in the class would also be a very welcome addition in the discussion.

Response: Ornidazole is an antimicrobial agent belonging to the 5-nitroimidazole class of compounds. Drug-induced liver injury has been documented with other antimicrobial agents, particularly albendazole, which has been associated with hepatotoxicity in several reported cases. Although metronidazole-induced liver injury occurs less frequently, isolated cases have been documented in the literature. The objective of this manuscript was to present a focused analysis of ornidazole-induced liver injury (OILI) as a distinct clinical entity. Through this investigation, we aimed to highlight the potential for drug-induced hepatotoxicity development in patients receiving ornidazole therapy.

Reviewer 2 Report

Comments and Suggestions for Authors

The manuscript title was “Ornidazole-Induced Liver Injury: Clinical Characterization of a Rare Adverse Reaction and Implications from a Multicenter Study”. The specific advice was as follow.

  1. Relevant investigation reports should be givenin the introduction.
  2. How to alleviate the liver damage caused by this drug?

Author Response

Reviewer 2:

Comments and Suggestions for Authors

The manuscript title was “Ornidazole-Induced Liver Injury: Clinical Characterization of a Rare Adverse Reaction and Implications from a Multicenter Study”. The specific advice was as follows.

  1. Relevant investigation reports should be given in the introduction.

Response: In our article, we planned to make a general introduction about toxic hepatitis and ornidazole in the introduction section. In the discussion section, we wanted to give detailed information about previous studies on OILI.

  1. How to alleviate the liver damage caused by this drug?

Response: Management of ornidazole-induced liver injury (OILI) follows the general principles of drug-induced hepatotoxicity, as no specific antidote exists. Most patients demonstrate improvement in liver biochemistry following ornidazole withdrawal. Nevertheless, severe hepatotoxicity may lead to fulminant hepatic failure, necessitating liver transplantation or resulting in fatal outcomes.

Reviewer 3 Report

Comments and Suggestions for Authors

Review of Manuscript-biomedicines-3716466: "Ornidazole-Induced Liver Injury: Clinical Characterization of a Rare Adverse Reaction and Implications from a Multicenter Study"

This study provides the most extensive clinical characterization of OILI to date, highlighting its potential severity, including risk of liver failure and death—and the importance of early recognition and drug discontinuation. The findings support considering ornidazole in the differential diagnosis of acute hepatitis, especially in endemic regions. I suggest addressing the points below to strengthen clinical applicability and discussion of limitations.

  1. All cases come from Turkey, where ornidazole use is frequent. Generalizability to other groups may be limited, as the presence of OILI elsewhere is unclear.
  2. Only 5.9% of patients were re-challenged with ornidazole, limiting data on recurrence or reproducibility of injury.
  3. Chronicity was rare (1%), but longer-term outcomes beyond six months are not detailed.
  4. Authors should briefly discuss potential mechanisms for OILI, including why some patients develop autoimmune features.
  5. The authors should consider the applicability of their findings in other countries and populations where the use of ornidazole is less common.
  6. If available, authors ought to present additional details on long-term follow-up, especially for patients with chronic DILI or transplantation.
  7. Authors should address study limitations such as retrospective design, underreporting, and lack of randomization in treatment techniques.
  8. While the paper's overall writing is effective, the authors could simplify several parts, particularly the methodology and results, to enhance clarity and conciseness.

Author Response

Reviewer 3:

Comments and Suggestions for Authors

Review of Manuscript-biomedicines-3716466: "Ornidazole-Induced Liver Injury: Clinical Characterization of a Rare Adverse Reaction and Implications from a Multicenter Study"

This study provides the most extensive clinical characterization of OILI to date, highlighting its potential severity, including risk of liver failure and death—and the importance of early recognition and drug discontinuation. The findings support considering ornidazole in the differential diagnosis of acute hepatitis, especially in endemic regions. I suggest addressing the points below to strengthen clinical applicability and discussion of limitations.

  1. All cases come from Turkey, where ornidazole use is frequent. Generalizability to other groups may be limited, as the presence of OILI elsewhere is unclear.

Response: Ornidazole, a synthetic 5-nitroimidazole antimicrobial agent, is indicated for the treatment of protozoal and anaerobic bacterial infections. Due to its broad-spectrum activity and favorable tolerability profile, ornidazole is extensively prescribed in several countries including Turkey, India, China, and various European nations. According to data from the Turkish Association of Liver Diseases, Drugs and Toxic Hepatitis Working Group, ornidazole represents the second most frequent cause of drug-induced liver injury in Turkey, following amoxicillin-clavulanic acid combination therapy. I am the secretary of this working group.

  1. Only 5.9% of patients were re-challenged with ornidazole, limiting data on recurrence or reproducibility of injury.

Response: Inadvertent rechallenge occurred when patients inadvertently resumed ornidazole therapy following their initial episode of drug-induced liver injury. All patients who underwent unintentional rechallenge developed recurrent hepatotoxicity upon ornidazole re-exposure.

  1. Chronicity was rare (1%), but longer-term outcomes beyond six months are not detailed.

Response: Follow-up was routinely discontinued following patient recovery in most cases. Additionally, incomplete long-term follow-up data beyond six months resulted from patient transitions to other healthcare facilities, which may limit the assessment of potential delayed complications.

  1. Authors should briefly discuss potential mechanisms for OILI, including why some patients develop autoimmune features.

Response: The pathogenetic mechanisms underlying autoimmune feature development in certain OILI patients remain incompletely understood, though genetic predisposition, individual immune system variability, and drug-specific immunological triggers may contribute to this phenomenon.

  1. The authors should consider the applicability of their findings in other countries and populations where the use of ornidazole is less common.

Response: Ornidazole is classified as a 5-nitroimidazole antimicrobial agent. This drug class, which includes metronidazole, tinidazole, and other derivatives, constitutes a cornerstone of therapy for anaerobic bacterial and protozoal infections worldwide. This comprehensive analysis advances our understanding of ornidazole-induced liver injury and provides clinically relevant data for healthcare practitioners managing patients with drug-induced hepatotoxicity.

  1. If available, authors ought to present additional details on long-term follow-up, especially for patients with chronic DILI or transplantation.

Response: The study endpoint was established as the final patient status following OILI management. Long-term post-transplantation outcomes were not systematically tracked, limiting our ability to assess transplant-related morbidity and mortality in this patient cohort.

  1. Authors should address study limitations such as retrospective design, underreporting, and lack of randomization in treatment techniques.

Response: The primary limitations of this study include its retrospective design, potential underreporting, and lack of standardization in treatment protocols.

  1. While the paper's overall writing is effective, the authors could simplify several parts, particularly the methodology and results, to enhance clarity and conciseness.

Response: The study protocol and results underwent rigorous internal review by all co-authors, with multiple revisions implemented to ensure methodological consistency and preserve data integrity. The revised manuscript underwent professional editing by MDPI Author Services to enhance clarity and readability.

Round 2

Reviewer 1 Report

Comments and Suggestions for Authors

The authors plan on reporting several aspects of the hepatotoxicity associated with ornidazole in separate manuscripts rather than submitting a comprehensive overview that details the histologic findings, etc. This plan seriously diminishes the quality of the current manuscript. It should not be difficult to at least add a few sentences in the introduction, results and/or  discussion section that describes the main indications for use of this agent in the Turkish population and the total usage of the drug (which they stated in their response to the reviewer's comments).  

I can accept the inclusion of the probable cases given the information provided by the authors - but their explanation should be included in the results. 

They nicely define the difference between autoimmune-like injury from the drug  (absence of relapse after steroid withdrawal)  and cases of induced autoimmune hepatitis (defined as a relapse after steroid withdrawal) in the response to the reviewer's comments - but fail to include this in the results. That should also be an easy fix. 

As far as the histology is concerned, a brief mention of the main pathologic features could be included here while saving the specifics for a separate manuscript. That way, the clinical signature of OILI would be more robust.  The small percentage of cases that were deemed to be cholestatic should have a more complete description included as the vast majority were hepatocellular. 

I understand that while the focus was on OILI,  a  discussion of the comparative hepatotoxicity of other members of the class should also be included in order to provide a more comprehensive overview, especially since ornidazole is not available in many countries, including the US.  If the authors have any insight into why it has not been approved in these countries, it would be useful to mention.  

Author Response

We thank you for your valuable comments and contributions to our manuscript.

The authors plan on reporting several aspects of the hepatotoxicity associated with ornidazole in separate manuscripts rather than submitting a comprehensive overview that details the histologic findings, etc. This plan seriously diminishes the quality of the current manuscript. It should not be difficult to at least add a few sentences in the introduction, results and/or discussion section that describes the main indications for use of this agent in the Turkish population and the total usage of the drug (which they stated in their response to the reviewer's comments).  

Response: In Turkey, ornidazole represents a commonly utilized antimicrobial agent in gynecological and dental practice. The medication is specifically prescribed for treating anaerobic infections within these medical specialties.

National prescription data for ornidazole in Turkey were not readily available for analysis. Nevertheless, a retrospective review of pharmaceutical records at our institution documented the prescription of 46,000 units during the 10-year study period.

Our future research focuses on comprehensive immunological evaluation of ornidazole-treated patients, comparing those who developed toxic hepatitis with unaffected controls. The study aims to identify potential immunological disparities and biomarkers between these distinct patient populations.

I can accept the inclusion of the probable cases given the information provided by the authors - but their explanation should be included in the results. 

Response: Study participants were selected based on Rucam causality assessment scores, including only cases classified as highly probable, probable, or possible for drug-induced hepatotoxicity.

This sentence was added to the method section.

They nicely define the difference between autoimmune-like injury from the drug (absence of relapse after steroid withdrawal) and cases of induced autoimmune hepatitis (defined as a relapse after steroid withdrawal) in the response to the reviewer's comments - but fail to include this in the results. That should also be an easy fix. 

Response: Patients with autoimmune hepatitis scores ≥6 were monitored; immunosuppression was tapered after one year. Recurrence classified the case as ornidazole-induced autoimmune hepatitis; absence of recurrence indicated Ornidazole-induced autoimmune-like hepatitis. It was based solely on a relapse after steroid withdrawal. No other criterion was used in this regard.

This sentence was added to the method section.

As far as the histology is concerned, a brief mention of the main pathologic features could be included here while saving the specifics for a separate manuscript. That way, the clinical signature of OILI would be more robust.  The small percentage of cases that were deemed to be cholestatic should have a more complete description included as the vast majority were hepatocellular. 

Response: The main pathological features were not presented in a table. However, a picture showing autoimmune features, a picture showing classic toxic hepatitis features, and a pathology picture of a patient in need of liver transplantation were included in the text as 3 figures.

I understand that while the focus was on OILI, a discussion of the comparative hepatotoxicity of other members of the class should also be included in order to provide a more comprehensive overview, especially since ornidazole is not available in many countries, including the US.  If the authors have any insight into why it has not been approved in these countries, it would be useful to mention.  

Response: Ornidazole is not approved in the United States and the European Union. No information was available on the U.S. Food and Drug Administration and European Medicines Agency websites.

Several factors may contribute to the non-approval of ornidazole in these jurisdictions: (1) safety considerations, particularly hepatotoxic potential; (2) availability of efficacious alternatives; (3) inadequate clinical evidence to meet regulatory standards; and (4) economic constraints related to regulatory submission costs.

Reviewer 2 Report

Comments and Suggestions for Authors

None.

Author Response

We thank you for your valuable comments and contributions to our manuscript.